# Inhibiting poly(ADP-ribosylation) improves axon regeneration

Alexandra B Byrne[1,2,3†], Rebecca D McWhirter[4,5], Yuichi Sekine[3,6], Stephen M Strittmatter[2,3,6], David M Miller III[4,5], Marc Hammarlund[1,2,3]*

[1]Department of Genetics, Yale University School of Medicine, New Haven, United States; [2]Department of Neuroscience, Yale University School of Medicine, New Haven, United States; [3]Program in Cellular Neuroscience, Neurodegeneration, and Repair, Yale University School of Medicine, New Haven, United States; [4]Department of Cell and Developmental Biology, Vanderbilt University, Nashville, United States; [5]Program in Neuroscience, Vanderbilt University, Nashville, United States; [6]Department of Neurology, Yale University School of Medicine, New Haven, United States

**Abstract** The ability of a neuron to regenerate its axon after injury depends in part on its intrinsic regenerative potential. Here, we identify novel intrinsic regulators of axon regeneration: poly(ADP-ribose) glycohodrolases (PARGs) and poly(ADP-ribose) polymerases (PARPs). PARGs, which remove poly(ADP-ribose) from proteins, act in injured *C. elegans* GABA motor neurons to enhance axon regeneration. PARG expression is regulated by DLK signaling, and PARGs mediate DLK function in enhancing axon regeneration. Conversely, PARPs, which add poly(ADP-ribose) to proteins, inhibit axon regeneration of both *C. elegans* GABA neurons and mammalian cortical neurons. Furthermore, chemical PARP inhibitors improve axon regeneration when administered after injury. Our results indicate that regulation of poly(ADP-ribose) levels is a critical function of the DLK regeneration pathway, that poly-(ADP ribosylation) inhibits axon regeneration across species, and that chemical inhibition of PARPs can elicit axon regeneration.

*For correspondence: marc.hammarlund@yale.edu

Present address: †Department of Neurobiology, University of Massachusetts Medical School, Worcester, United States

## Introduction

Unlike damaged peripheral nerves, the central nervous system does not successfully regenerate after injury. Failure to regenerate has been attributed to two components of the regeneration response: intrinsic and extrinsic factors. While extrinsic inhibitory factors such as the glial microenvironment can be modulated with some success, regeneration potential is still substantially hindered, providing evidence that intrinsic factors play a significant role in modulating the ability of an axon to regenerate (*Richardson et al., 1980*; *Neumann and Woolf, 1999*; *GrandPré et al., 2000*; *Fournier et al., 2001*; *Qiu et al., 2002*; *Yiu and He, 2006*; *Park et al., 2008*; *Smith et al., 2009*; *Wang et al., 2011*). Developing an understanding of the intrinsic mechanisms that regulate regeneration will provide insight into the treatment of neurological injury and disease.

DLK-1 (Dual Leucine Zipper Kinase) is a mitogen activated protein kinase kinase kinase (MAPKKK) identified in *C. elegans* that functions intrinsically to regulate regeneration of adult axons in the central and peripheral nervous systems across species, including flies and mammals (*Hammarlund et al., 2009*; *Yan et al., 2009*; *Xiong et al., 2010*; *Shin et al., 2012*; *Wang et al., 2013*; *Watkins et al., 2013*; *Byrne et al., 2014*). Activation of *dlk-1* enhances axon regeneration and loss of *dlk-1* function inhibits axon regeneration in young and aged animals (*Hammarlund et al., 2009*; *Yan et al., 2009*; *Byrne et al., 2014*). In worms, flies, and mice, the function of DLK signaling

**eLife digest** Neurons carry information around the body along slender projections known as axons. An injury that crushes or cuts an axon can lead to permanent disability if the axon fails to regenerate. While some damaged neurons in the body can repair themselves, typically those present in the brain and spinal cord cannot regenerate successfully after injury.

The ability of a neuron to regenerate its axon depends in part on factors present inside the neuron itself. By understanding how these internal mechanisms regulate axon regeneration, researchers hope to develop new ways to boost the repair of damaged neurons.

A protein called DLK acts inside neurons to promote regeneration of injured axons across a range of species including worms and mammals. In the absence of DLK, regeneration is impaired. The DLK signaling pathway is activated in damaged neurons and is thought to promote repair by altering the activity of genes and proteins that control the regeneration process.

Byrne et al. have now identified genes that are activated by the DLK signaling pathway in the roundworm, *Caenorhabditis elegans*. The experiments show that DLK signaling increases the activity of genes encoding enzymes known as PARGs, which in turn enhance axon regeneration. PARG enzymes remove chain-like molecules called poly(ADP-ribose) that are attached to target proteins.

Further experiments showed that other enzymes known as PARPs, which add the poly(ADP-ribose) markers to proteins, act to inhibit axon regeneration in both *Caenorhabditis elegans* and in injured neurons from mice. Consistent with this, Byrne et al. found that drugs that inhibit PARP enzymes improved axon regeneration when they were given to *C. elegans* with injured neurons. These results suggest that a critical role of the DLK signaling pathway is to regulate poly(ADP-ribose) levels and that reducing the amount of poly(ADP-ribose) added to proteins can promote axon regeneration.

The next step is to understand exactly how poly(ADP-ribose) regulates axon regeneration and to identify the other factors – besides poly(ADP-ribose), PARGs and PARPs – that act downstream of DLK signalling to regulate regeneration.

in regeneration depends on gene transcription (*Xiong et al., 2010*; *Shin et al., 2012*; *Yan and Jin, 2012*; *Watkins et al., 2013*; *Stone et al., 2014*). These data suggest that specific targets of DLK transcriptional regulation may mediate the ability of DLK signaling to promote regeneration. Further, these targets may identify novel aspects of the cell biology of axon regeneration. Finally, modulation of these targets might increase the intrinsic regenerative potential of injured axons.

## Results and discussion

To identify targets of DLK transcriptional regulation in neurons, we took advantage of a recently developed method that uses FACS to isolate *C. elegans* neurons and compare their gene expression profiles (*Spencer et al., 2014*). We sorted GABA motor neurons from animals with activated DLK signaling (*dlk-1(OE)*, conferred by overexpression of DLK-1L [*Hammarlund et al., 2009*; *Yan and Jin, 2012*]) and compared them to *wild-type* GABA neurons. To control for potential off-target effects of DLK activation, we also analyzed neurons that contained both *dlk-1(OE)* and a loss of function mutation in *pmk-3,* the MAP kinase at the end of the canonical DLK signaling pathway (*Nakata et al., 2005*; *Hammarlund et al., 2009*; *Yan and Jin, 2012*). RNA sequencing and analysis suggested the *parg* genes as candidates for further evaluation. The gene *parg-2* (poly(ADP-ribose) glycohydrolase-2) was significantly upregulated in neurons with activated DLK signaling (187-fold upregulated in *dlk-1(OE)* vs *wild type*, p<0.01, two-way ANOVA, Bonferroni post-test) (*Figure 1A*). Further, examination of RNA-Seq results for the *parg*-2 paralog, *parg-1,* detected a 2.5-fold increase in the *dlk-1(OE)* background compared to *wild type* (p<0.05, two-way ANOVA, Bonferroni post-test, *Figure 1A*) (See Materials and methods). Elevated expression of *parg-1* and *parg-2* by DLK signaling depended on the canonical DLK MAP kinase pathway since up-regulation was eliminated in neurons that over-expressed *dlk-1* but lacked its downstream effector *pmk-3* (*Figure 1A*). These data suggested that regulation of PARG function might be a major effect of DLK signaling. Overall, up-regulation (>two fold, p<0.05) (See Materials and methods) of gene expression by DLK signaling was

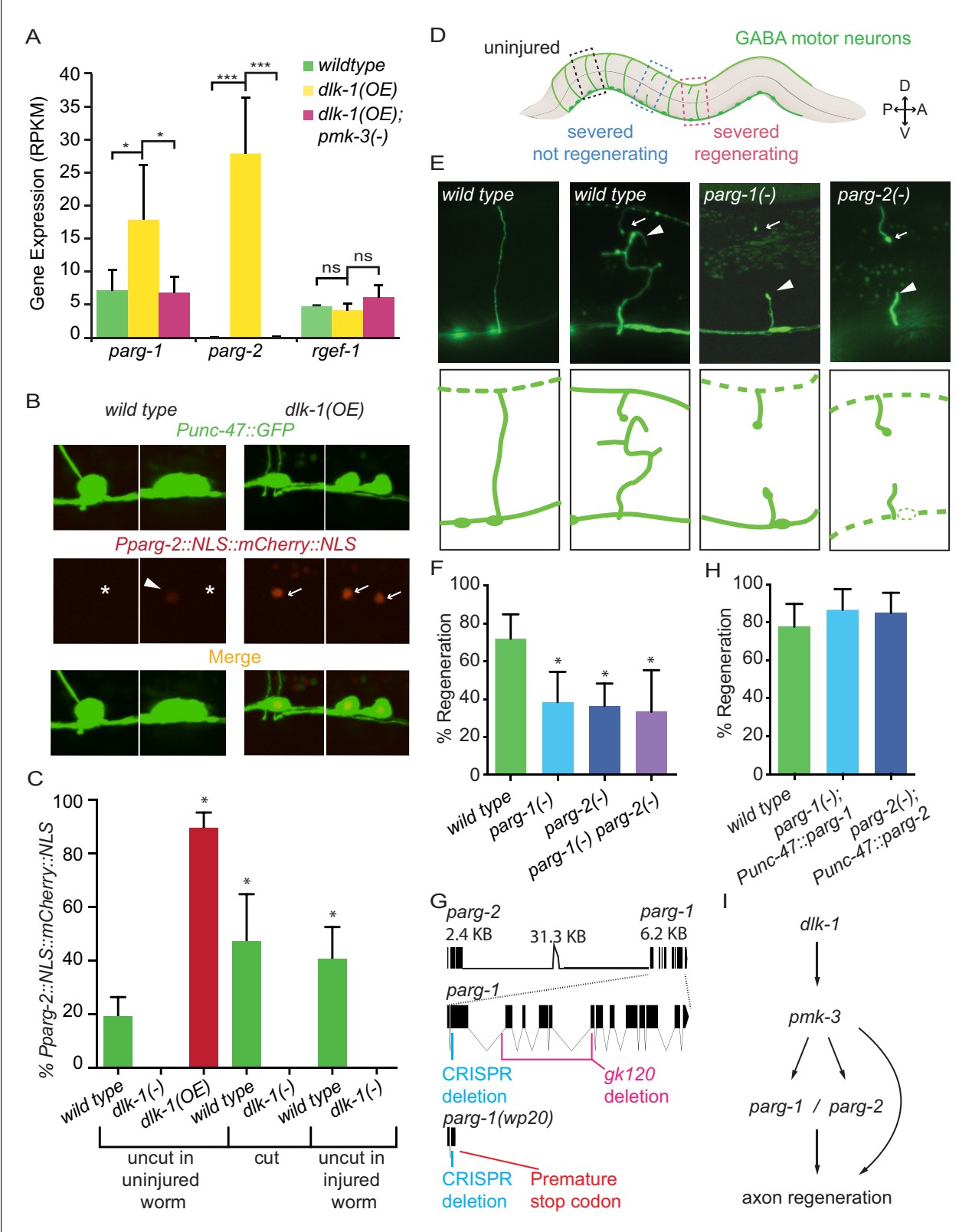

**Figure 1.** PARG genes regulate axon regeneration. (**A**) *dlk-1* overexpression upregulates *parg-1* and *parg-2* expression levels in GABA neurons. The upregulation is suppressed by loss of *pmk-3* function. *rgef-1* (a pan-neuronal Ras nucleotide exchange factor) expression levels are not affected by manipulations of the *dlk-1* pathway (*p<0.05, ***p<0.01, two-way ANOVA, Bonferroni post-test). (**B–C**) *dlk-1* regulates expression of nuclear-localized mCherry driven by the *parg-2* promoter. *Pparg-2::NLS::mCherry::NLS* was observed (arrows) in 90% of nuclei of GABA neurons in *dlk-1(OE)* animals and

*Figure 1 continued on next page*

*Figure 1 continued*

in 19% of GABA neurons in *wild type* animals (asterisks). GABA neurons express GABA neuron-specific GFP marker, *Punc-47::GFP*. (C) *parg-2* expression was significantly increased in both severed axons and neighboring uncut axons relative to axons in uninjured *wild type* animals. *parg-2* was not expressed in *dlk-1(lf)* axons, whether severed or intact. (*p<0.05, Fisher's exact test, relative to wild type, n = 111, 20, 115, 34, 18, 69, 36). (D) The GABA motor nervous system of *C. elegans*. GFP-labeled axons were severed with a pulsed laser at the midline (dark brown line) and scored for regeneration. (E) Representative micrographs of uninjured *wild type*, severed *wild type*, severed *parg-1(-)*, and severed *parg-2(-)* GABA axons. Each carry the *oxIs12* transgene which drives GFP expression in GABA neurons. Arrowheads and arrows indicate proximal and distal stumps, respectively. (F) Axon regeneration is significantly reduced in *parg-1(-)*, *parg-2(-)* and *parg-1(-) parg-2(-)* mutants compared to wild type animals (*p<0.05, Fisher's exact test, relative to wild type, n = 50, 39, 67, 21). (G) *parg-1* and *parg-2* are closely linked on chromosome IV, making construction of a double mutant difficult. To create a double *parg-1 parg-2* mutant, *parg-1* was mutated with CRISPR in a *parg-2(lf)* background. The resulting frameshift mutation (*wp20*) truncates PARP-1 earlier than the canonical *gk120* deletion allele. (H) Expression of *parg-1* or *parg-2* in GABA motor neurons rescued axon regeneration in *parg-1(lf)* and *parg-2(lf)* mutants, respectively. (I) Model of PARG function in axon regeneration.

observed for only 1.9% of coding genes (386 out of 20,375 protein coding genes assayed); expression of most genes was not affected by DLK signaling. For example, expression levels of the pan-neuronal control gene *rgef-1* (a ras nucleotide exchange factor) were not altered in either mutant background (*Figure 1A*). Together, our data indicate *parg* expression is regulated by DLK signaling in GABA neurons.

To further test whether DLK regulates *parg* expression in GABA neurons, we built a reporter construct that expresses nuclear-localized mCherry driven by the *parg-2* promoter. At low magnification (4X), mCherry was only detected in *dlk-1(OE)* animals (20/20 *dlk-1OE* animals, 0/20 wild type animals). At high magnification (40X), mCherry was seen in 90% of GABA neurons in *dlk-1(OE)* animals and 19% of GABA neurons in control animals (*Figure 1B,C*) (p<0.0001, Fisher's exact test). mCherry was not detected in *dlk-1(lf)* animals, which lack DLK-1. Therefore, *parg-2* expression is dependent on *dlk-1*, even in intact, uninjured axons.

Next, we tested the effect of axon injury on *dlk-1*-dependent *parg* expression (*Figure 1B,C*). Approximately 10 hr post-axotomy, *parg-2* expression was significantly elevated in cut GABA axons relative to *parg-2* expression in GABA axons of uninjured animals (47% vs 19%, p=0.0028, Fisher's exact test). Further, by examining neighboring uncut GABA axons, we found that *parg-2* expression also increased in uninjured neurons to equivalent levels (47% vs 41%, p=0.6722, Fisher's exact test). The increase in *parg-2* expression in response to injury is entirely dependent on *dlk-1*, as *parg-2* expression was not seen in cut or uncut axons in injured *dlk-1(lf)* animals (p=0.0003 and p=0.0001, relative to cut axons and uncut axons in injured wild type animals). Thus, *parg-2* expression is upregulated in injured neurons and their neighbors, and dependent on *dlk-1*.

Poly(ADP-ribose) glycohydrolases (PARGs) catalyze dePARylation: the removal of the post-translational modification poly(ADP-ribose) (PAR) from target proteins (*Miwa and Sugimura, 1971*; *Althaus and Richter, 1987*). The *parg-1* and *parg-2* genes encode the only two PARGs in the *C. elegans* genome (*Gagnon et al., 2002*). We determined the function of *parg-1* and *parg-2* in axon regeneration by assessing regrowth after single neuron laser axotomy in GABA neurons (*Byrne et al., 2011*) (*Figure 1D*). Loss of either *parg-1* or *parg-2* reduced axon regeneration to approximately half of normal levels: only 39% and 36% of axons regenerated in *parg-1* and *parg-2* mutants, respectively, while 70% of axons regenerated in control animals (*Figure 1E,F*). Therefore, *parg-1* and *parg-2* regulate axon regeneration.

The *parg-1* and *parg-2* genes are closely linked on chromosome IV, complicating generation of a double mutant. To assess whether complete elimination of PARG activity could further reduce regeneration, we used a CRISPR-Cas9 approach (*Friedland et al., 2013*) to mutate *parg-1* in the *parg-2(lf)* background (*Figure 1G*). The resulting *parg-1(lf); parg-2(lf)* double mutant was viable and displayed wild-type morphology and behavior, indicating PARG function is not essential. Axon regeneration in *parg-1(lf); parg-2(lf)* animals was similar to axon regeneration in either *parg-1* or *parg-2* mutant animals (*Figure 1F*). Thus, PARG activity is required for normal axon regeneration, but some regeneration occurs even in animals that completely lack *parg*.

To test whether the *parg* genes act within GABA neurons to regulate axon regeneration, we reintroduced *parg-1* or *parg-2* specifically in GABA neurons (using the *unc-47* promoter) of *parg-1(lf)* or *parg-2(lf)* mutants, respectively, and assessed regeneration. We found that 87% and 85% of injured

axons regenerated in *parg-1* and *parg-2* worms whose GABA neurons had restored PARG expression (*Figure 1H*). We conclude cell-intrinsic PARG function is required for axon regeneration of GABA neurons (*Figure 1I*).

Cellular levels of PARylation are determined by the balance between the activity of PARGs, which remove PAR, and the activity of poly (ADP-ribose) polymerases (PARPs), which transfer PAR onto target proteins (*Schreiber et al., 2006*; *Gibson and Kraus, 2012*). Thus, axon regeneration defects in *parg-1* and *parg-2* mutants (*Figure 1F*) could be due to accumulation of PAR. To test this hypothesis, we analyzed regeneration in animals with reduced PAR. The *C. elegans* genome contains two PARP homologs, *parp-1* and *parp-2* (*Gagnon et al., 2002*). We found that mutation of either *parp-1* or *parp-2* increased axon regeneration relative to control animals: 92% and 90% of axons regenerated in *parp-1* and *parp-2* mutants, respectively, while 76% of axons in controls regenerated (*Figure 2A,B*). Regenerating axons in these assays include all those that initiate a migrating growth cone after injury. To determine whether axons in PARP mutants are capable of sustained growth toward their original target, we assessed ability to extend towards the dorsal nerve cord (*Figure 2C*). We found that 56% and 53% of regenerating axons in *parp-1* and *parp-2* mutants, respectively, regrew at least 3/4 of the distance between the ventral and the dorsal nerve cords compared to only 26% of regenerating axons in controls (*Figure 2D*). Moreover, some *parp-2* mutants sprouted new axons from the cell body (*Figure 2E*). The opposing effects of PARGs and PARPs on poly(ADP)-ribose and on axon regeneration indicate that PARylation is a critical determinant of regenerative potential.

PARG and PARP function are well-conserved between *C.elegans* and mammals (*Gagnon et al., 2002*; *St-Laurent et al., 2007*). Therefore, we hypothesized that blocking PARP function might be sufficient to improve regeneration of mammalian CNS neurons. To test this hypothesis, we assessed the effect of PARP knockdown on mouse cortical neuron regeneration. We cultured primary cortical mouse neurons in 96-well plates (*Huebner et al., 2011*). We subsequently added lentiviral control or one of two unique PARP1 shRNAs at three days in vitro (DIV), and injured the neurons with a custom pin-replicator five days later (*Huebner et al., 2011*). Three days after injury, we fixed the neurons and assessed regeneration. We found that axons exposed to PARP1 shRNA regenerated significantly better than axons exposed to control shRNA (*Figure 2F,G*). To confirm the shRNA clones targeted PARP1, we performed western blots on cortical neurons exposed to the negative control shRNA or to the two unique shRNA that target PARP1. In each case PARP was detected in the insoluble fraction and not in the lysate, in agreement with previously reported localization to the nucleus (reviewed in *Bai, 2015*). PARP was significantly reduced in neurons exposed to either of the PARP1-targeting shRNAs compared to negative control shRNA (*Figure 2H,I*). PARP levels were normalized to actin levels in each sample of neurons. Therefore, PARP-1 and PARylation are conserved inhibitors of axon regeneration after injury, and reducing their function improves axon regeneration across species.

Having established that PARGs are novel regulators of axon regeneration, we sought to determine the extent to which DLK function is mediated by PARGs. We assessed regeneration in animals with activated DLK signaling (*dlk-1(OE)*), but lacking both *parg-1* and *parg-2*. We found that loss of both *parg-1* and *parg-2* function reduced regeneration in the *dlk-1(OE)* background, just as loss of *parg-1* and *parg-2* reduces regeneration in animals with wild type levels of DLK signaling (*Figure 2J* and *Figure 2—figure supplement 1*). In both cases, regeneration is reduced but not eliminated, and the amount of regeneration that remains is higher than complete loss of *dlk-1* signaling (0% regeneration) or loss of the downstream *pmk-3* in the *dlk-1(OE)* background (7% regeneration) (*Nakata et al., 2005*; *Hammarlund et al., 2009*). Thus, DLK-dependent regeneration depends in part on *parg-1* and *parg-2*.

In addition to controlling axon regeneration, DLK signaling regulates presynaptic development (*Nakata et al., 2005*). To test whether presynaptic development is regulated by PARylation, we quantified synapses at the GABA neuromuscular junction with the pre-synaptic reporter *hpls3*. The *hpls3* reporter expresses GFP tagged SYD-2 (alpha-liprin) in presynaptic active zones of GABA motor neurons (*Zhen and Jin, 1999*). In control animals, SYD-2::GFP is distributed in a punctate pattern at regularly interspaced intervals along the dorsal nerve cord (*Figure 3*) (*Yeh et al., 2005*). Loss of all PARG activity did not affect synapse morphology. Increased DLK signaling in *dlk-1(OE)* animals causes synapse morphology defects (*Figure 3*) (*Nakata et al., 2005*). In *dlk-1(OE)* animals, SYD-2::GFP is diffuse along the dorsal nerve cord, which increases the average baseline fluorescence along

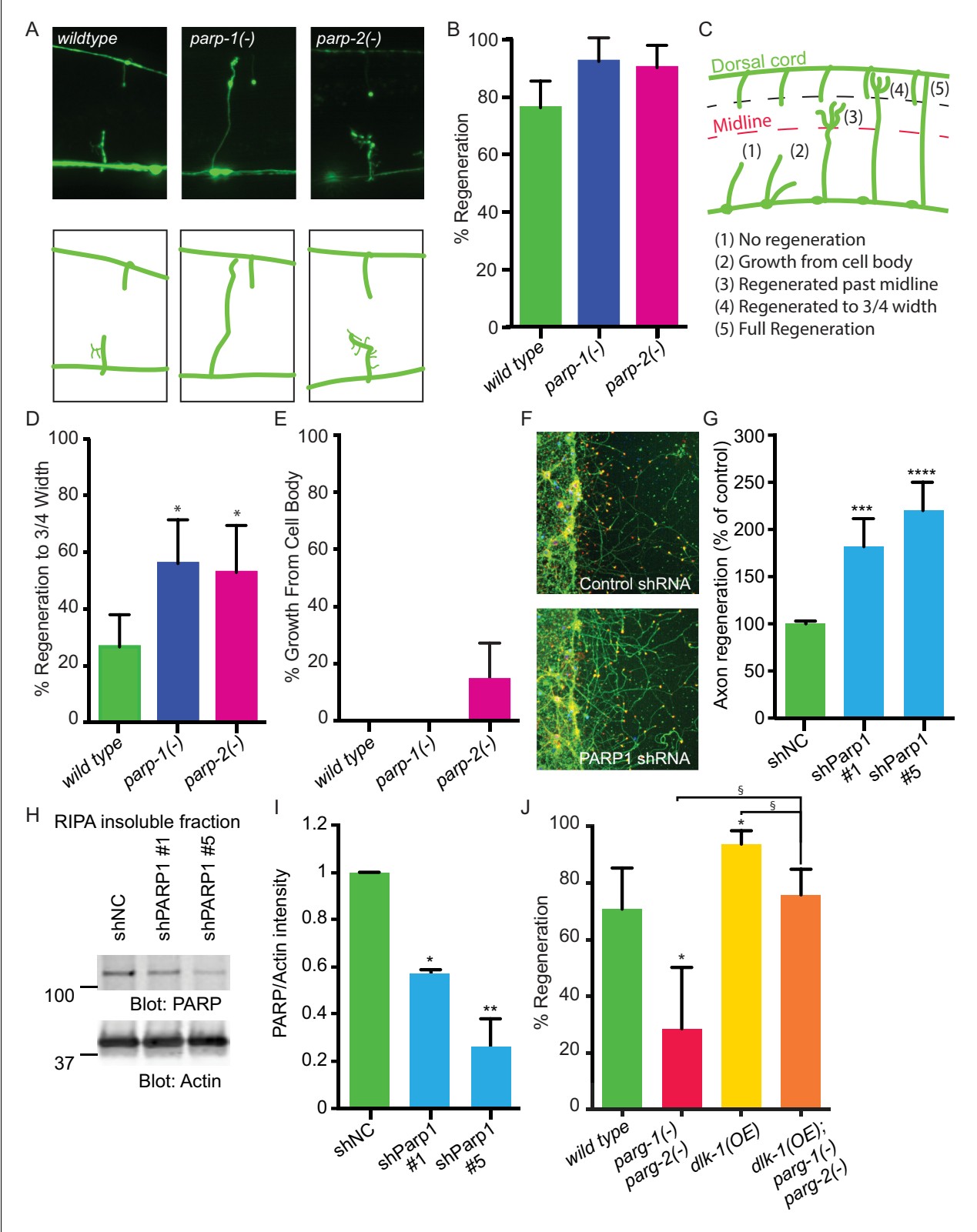

**Figure 2.** PARPs inhibit axon regeneration. (**A**) Representative micrographs of severed wild type, *parp-1(lf)*, and *parp-2(lf)* GABA motor neurons. (**B**) Axon regeneration in *parp-1(lf)* and *parp-2(lf)* mutants compared to wild type animals (*p<0.05, Fisher's exact test, n = 84, 26, 61). (**C**) Cut axons (1) are scored for the distance they extend towards their targets in the dorsal nerve cord (2, 3, 4, 5). (**D**) Axon regeneration to at least 3/4 of the distance to the dorsal cord (4) is significantly increased in *parp-1(lf)* and *parp-2(lf)* mutants relative to *wild type* animals (*p<0.01, Fisher's exact test, n = 61, 43, 38). (**E**) *Figure 2 continued on next page*

*Figure 2 continued*

Axon regeneration from the cell body (2) is seen in *parp-2(lf)* mutants (n = 47, 32, 34). (**F**) Representative micrographs of injured cortical neurons exposed to negative control shRNA or PARP1 shRNA. (**G**) Axon regeneration is increased in murine cortical neurons lacking PARP1 (***p<0.001, ****p<0.0001, Anova with Tukey's multiple comparisons test, n = 108, 8, 8). Axon regeneration was measured in injured cortical neurons exposed to non-coding negative control shRNA (shNC) or either of two unique PARP1 shRNAs. (**H, I**) Exposure to either shPARP significantly reduced PARP levels in cortical neurons relative to PARP levels in cortical neurons exposed to negative control (shNC) lentivirus (*p<0.05, **p<0.005, Anova with Tukey's multiple comparisons test). (**J**) *parg-1* and *parg-2* loss of function incompletely suppress the increase in regeneration conferred by *dlk-1(OE)* (*p<0.05, relative to *wild type*, §p<0.05, relative to *dlk-1(OE)*, Fisher's exact test, n = 24, 21, 48, 62), indicating the PARGs regulate regeneration downstream of *dlk-1* with at least one parallel pathway.

The following figure supplement is available for figure 2:

**Figure supplement 1.** Detailed characterization of regeneration in *dlk-1(OE)* and *parg-2(-) parg-2(-)* mutants.

a line scan (*Figure 3*). However, loss of PARG activity did not suppress these defects. Together, these data indicate that in contrast to its role in axon regeneration, PARylation does not regulate synapse formation, even when DLK signaling is activated.

Axon injury triggers an acute response that includes activation of DLK signaling (*Yan and Jin, 2012*). PARylation is a short-lived modification, and PAR levels are normally maintained by the continuous activity of PARP and PARG proteins (*Schreiber et al., 2006*; *Gibson and Kraus, 2012*). These data suggest a model in which increased PARG expression downstream of DLK signaling acutely reduces PAR levels in response to axon injury, thereby facilitating regeneration. We hypothesized that acute reduction of PAR levels by inhibition of PARP might also increase axon regeneration after injury, potentially similar to increased regeneration in PARP mutants (*Figure 2*). Multiple chemical PARP inhibitors are currently in preclinical and clinical trials for indications including cancer therapy and stroke (*Ford and Lee, 2011*; *Anwar et al., 2015*). We found that treatment with chemical PARP inhibitors after injury resulted in significantly enhanced axon regeneration in vivo in *C. elegans* GABA neurons and in vitro in murine cortical neurons (*Figure 4A–C*, and *Figure 4—figure supplement 1*). Drug treatment post-injury also improved behavioral recovery, demonstrating that enhanced regeneration after PARP inhibition results in functional reconnection (*Figure 4D,E*). Thus, acute poly(ADP-ribose) levels determine the response of neurons to axon injury, and inhibition of PARP after injury is sufficient to improve regeneration.

Together, our findings suggest that regulation of PARylation is an important component of the DLK pathway role in the axon regeneration mechanism. Multiple lines of evidence suggest the *parg* genes are transcriptionally regulated downstream of DLK signaling to promote regeneration. First, we find both of the *parg* genes are upregulated in animals with activated DLK signaling. Second, we find that endogenous *dlk-1* signaling drives *parg-2* expression in an injury-dependent manner. Third, we find that loss of the *parg* genes reduces regeneration, both in animals with endogenous levels of *dlk-1* activity and in animals with elevated DLK-1 signaling. These findings suggest a linear model in which DLK signaling induces *parg* expression, which in turn facilitates regeneration by removing PAR.

In addition to regulating PARylation, our data indicate that DLK signaling regulates regeneration upstream of multiple effectors. Although PARylation has a strong effect on regeneration in both animals with wild-type levels of *dlk-1* and in animals that overexpress *dlk-1*, some GABA axons in *parg-1(lf) parg-2(lf)* mutants are still able to regenerate (*Figure 1F*), indicating high levels of PAR do not completely prevent DLK-mediated axon regeneration. By contrast, GABA axons do not regenerate in animals lacking DLK (*Hammarlund et al., 2009*; *Yan et al., 2009*). These data suggest that DLK activity has other functions besides regulating PARylation. Some of these functions may be mediated by other transcriptional outputs of DLK signaling (*Watkins et al., 2013*). Understanding these factors, as well as understanding the cellular effects of PAR on regeneration, await further study.

Besides shedding light on functional outputs of DLK signaling, our findings identify a novel pathway, involving control of poly(ADP-ribose) levels, that regulates axon regeneration (*Figure 4F*). Specifically, we find that PARG and PARP activity regulate the acute response of neurons to axon injury, and that chemical PARP inhibition after injury is sufficient to improve regeneration. The lack of additive phenotype in the double loss of function *parg-1(-); parg-2(-)* mutant suggests the two *parg*

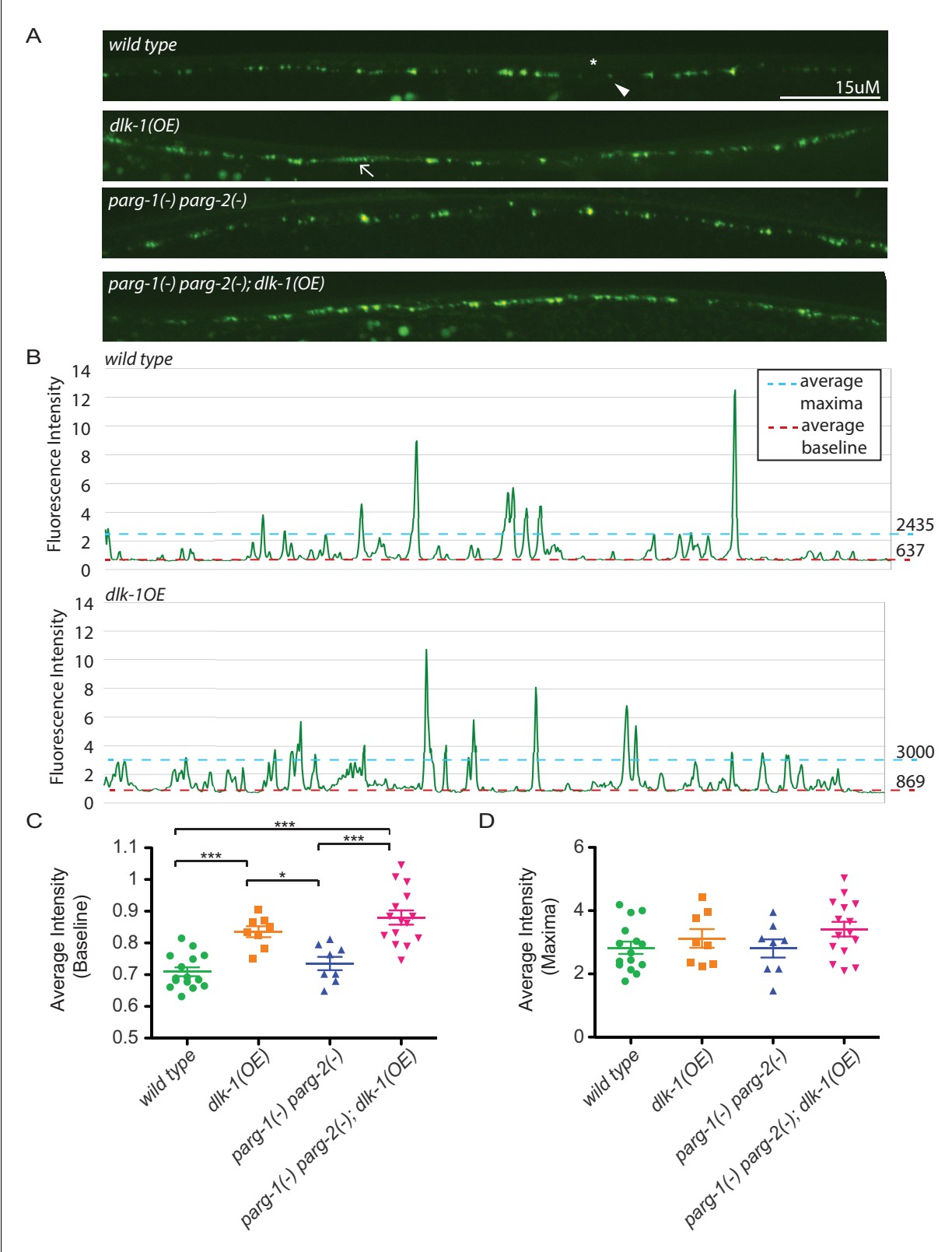

**Figure 3.** Loss of *parg-1* and *parg-2* function does not suppress mislocalizion of presynaptic active zones caused by *dlk-1* overexpression. (**A**) Dorsal nerve cords of wild type, *dlk-1(OE)*, *parg-1(lf) parg-2(lf)*, and *dlk-1(OE); parg-1(lf) parg-2(lf)* animals. All animals express the presynaptic active zone marker SYD-2::GFP in their GABA neurons. SYD-2::GFP is expressed in discrete puncta (arrowhead) in wild type animals and is not expressed continuously along the dorsal cord (asterisk). Conversely, SYD-2::GFP is expressed in a diffuse pattern (arrow) in *dlk-1(OE)* animals. (**B**) Average maxima

*Figure 3 continued on next page*

*Figure 3 continued*

and average baseline fluorescence are calculated along line scans of each dorsal cord and represented in (C). (C) *dlk-1* overexpression disrupts SYD-2:: GFP expression in GABA neurons and results in higher baseline fluorescence compared to wild type. Loss of parg function does not affect localization of SYD-2::GFP nor does it suppress the mislocalization caused by *dlk-1(OE)* (*p<0.05, ***p<0.01, multiple ANOVA, Bonferroni post-test). (D) There are no significant differences in average maxima between genotypes. Sample size is 15, 8, 8, and 15 animals for *wild type, dlk-1(OE), parg-1(lf) parg-2(lf)*, and *dlk-1(OE); parg-1(lf) parg-2(lf)* animals, respectively.

genes are not partially redundant. Rather, *parg-1* and *parg-2* may function together, for example as part of a complex. In *Arabidopsis thaliana*, the two *PARG* homologs physically interact (*Song et al., 2015*), suggesting the *PARG* homologs may function coordinately. Alternatively, concerted action of both PARGs may be required to maintain PAR levels below a threshold. In this model, loss of either single PARG results in a sufficient PAR increase to block regeneration, but increasing PAR beyond this threshold does not further reduce regeneration.

Previous investigations of the relationship between the two *C. elegans parg* homologs have been complicated by the physical proximity of the two genes in the genome. As a result, double loss of function mutants have been generated using RNAi. Since RNAi can result in incomplete knockdown of target genes, it has been difficult to determine the functional redundancy of the two genes using this approach. The *parg-1; parg-2* mutant described here may be useful for further characterization of animals that completely lack PARG function.

In vivo, injured mammalian axons must overcome extrinsic growth inhibition to regenerate. PARP1 is upregulated in murine cortical neurons exposed to inhibitory growth molecules (myelin-associated glycoprotein, Nogo-A, Chondroitin sulfate proteoglycans) in vitro and in crushed optic nerves in vivo. Moreover, inhibiting PARP1 promotes neurite outgrowth on inhibitory substrates in vitro (*Brochier et al., 2015*). Since PARPs and PARGs have contrasting effects on PAR levels, NAD+ levels, which are substrates of PAR (*Bai, 2015*), and on axon regeneration, we conclude the balance between PARP and PARG function regulates axon regeneration, and present the hypothesis the PARG-PARP balance may determine axon regeneration by regulating PAR levels or by regulating NAD+ levels. Finally, the conservation of the role of PARP in mammalian axon regeneration may have important implications for nerve repair following injury or disease.

## Materials and methods

### *C. elegans* strains

Strains were maintained as previously described at 20°C (*Brenner, 1974*). Some strains were provided by the CGC, which is funded by NIH Office of Research Infrastructure Programs (P40 OD010440). Specific mutations analyzed: *parg-1(gk120), parg-2(ok980), parp-1(ok988), parp-2 (ok344), wpIs9[Punc-47:DLK-1mini-GFP, ccGFP], pmk-3(ok169), hpIs3[punc-25::SYD-2::GFP; lin-15+]*. To visualize GABA neurons in regeneration assays, mutants were crossed into the *oxIs12 [Punc-47: GFP, lin-15+]* background. XE1347 *wpIs39[Punc-47:mCherry]*, XE1551 *wpIs9[Punc-47:DLK-1mini-GFP, ccGFP]; wpIs39[Punc-47:mCherry]*, and XE1552 *wpIs9[Punc-47:DLK-1mini-GFP, ccGFP]; pmk-3 (ok169); wpIs39[Punc-47:mCherry]* were analysed by RNA Seq.

### RNA-Seq

The SeqCel method was used to generate RNA Seq profiles of larval GABA neurons (*Spencer et al., 2014*). Briefly, L4 stage larvae were dissociated and *punc-47::mCherry*-labeled GABA neurons were isolated by FACS (BD FACSaria) from wild-type (XE1347), *dlk-1(OE)* (XE1551) and *dlk-1(OE); pmk-3 (ok169)* (XE1552) strains; dead and damaged cells were excluded by DAPI staining. Experiments were performed in triplicate for each genotype. For RNA-Seq analysis, total RNA (5–10 ng) was amplified by SMARTer cDNA synthesis (Clonetech) and libraries sequenced (PE-100) using the HiSeq 2500 system (Illumina). RNA-Seq data were analyzed with CLC Genomics Workbench software (Qiagen). A global comparison (EDGE test) (*Robinson et al., 2010*) of wild-type vs *dlk-1(OE)* GABA neuron RNA-Seq data sets detected 386 transcripts that are significantly up-regulated ($\geq$2 fold, p<0.05) in the *dlk-1(OE)* GABA neuron profile. A comprehensive analysis of these data sets will be presented

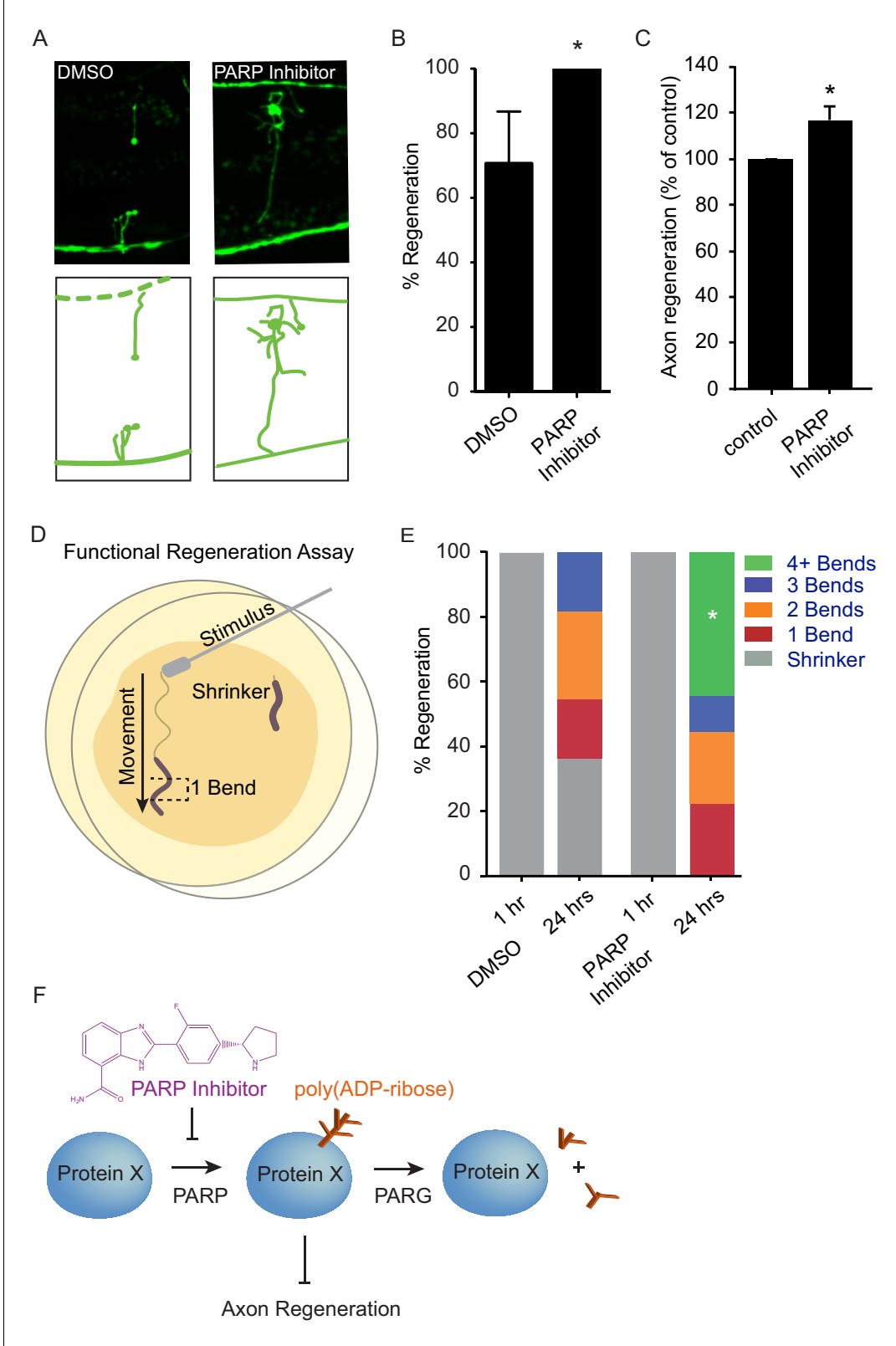

**Figure 4.** Chemical PARP inhibition enhances axon regeneration post-injury. (**A**) Micrographs of regenerating axons placed on plates containing DMS0 or PARP inhibitor (A966492, Selleckchem, 100 μM) immediately after surgery. (**B**) Acute chemical inhibition of PARP function enhances regeneration (*p<0.05, Fisher's exact test, n = 34 and 19 axons severed in animals exposed to DMSO or PARP inhibitor, respectively). (**C**) Axon regeneration is increased in murine cortical neurons exposed to chemical PARP inhibitor A966492 (*p=0.0149, Student's t-test, n = 10). (**D**) To assess functional

*Figure 4 continued on next page*

*Figure 4 continued*

regeneration, all GABA neurons were severed and animals were assessed for their ability to reverse in response to a touch on the nose from a platinum wire. (E) One hour after all GABA neurons are severed, animals are incapable of reversing in response to a touch on the nose (shrinker). As functional connections are regenerated, animals recovered on PARP inhibitor displayed more backward movement than those recovered on DMSO (measured as number of body bends). Significantly more animals on PARP inhibitors recovered wild type function (4+ body bends, *p<0.05, Fisher's exact test, n = 11 and 9 animals exposed to DMSO or PARP inhibitor, respectively). (F) The balance between PARP and PARG regulates axon regeneration and is altered by chemical PARP inhibitors.

The following figure supplement is available for figure 4:

**Figure supplement 1.** Chemical PARP inhibitors have different effects on axon regeneration.

elsewhere. The *parg-2* transcript was significantly enriched (187x, p=1.6 e$^{-14}$) in the *dlk-1(OE)* profile. The *parg-1* transcript was 2.5 fold elevated in *dlk-1(OE)* but was excluded from this initial analysis due to a conservative p-value correction for multiple testing. We identified *parg-1* as a likely false negative in this global analysis due to statistically significant elevation of the *parg-1* signal in a direct comparison with wild-type and *pmk-3(ok169)* (XE1552) (see *Figure 1A*).

## Expression analysis

*Pparg-2::NLS::mCherry::NLS* was constructed by combining Gateway plasmids encoding the *parg-2* promoter sequence (obtained from GE Dharmacon Promoterome collection), *NLS::mCherry::NLS* coding sequence, the *unc-54* UTR sequence, and *pCFJ150. Pparg-2::NLS::mCherry::NLS* was injected along with the *Pmyo-2::GFP* co-injection marker (expressed in the pharynx) into *wpIs9[Punc-47:DLK-1mini-GFP, ccGFP]; oxIs12[Punc-47::GFP]* animals. *wpIs9* was outcrossed from transgenic lines with wild type N2 males. mCherry expression was compared between animals carrying the same extra-chromosomal array in the presence or absence of *wpIs9*. The posterior 7 VD/DD GABA neurons in 10 worms were analyzed for each genotype. Expression was analysed with an Olympus DSU mounted on an Olympus BX61 microscope, Andor Neo sCMOS camera, and Lumen light source. Error bars represent 95% confidence intervals. Significance is indicated with an asterisk (p<0.0001, Fisher exact test).

## Axotomy experiments

Axotomy experiments were carried out as previously described (*Byrne et al., 2011*). Post-axotomy images were acquired with an Olympus DSU mounted on an Olympus BX61 microscope, Andor Neo sCMOS camera, and Lumen light source. Error bars represent 95% confidence intervals. Significance is indicated with an asterisk (p<0.01, Fisher exact test).

## CRISPR

The double *parg-1(wp20) parg-2(ok980)* mutant was created by injecting sgRNA targeting *parg-1* sequence: aaagactacgaagactatgt and Cas9 into *parg-2(ok980)* animals. The resulting deletion of the 20th and 21st base pairs of the second *parg-1* exon is a frameshift mutation that creates a truncated protein.

## Transgenics

*Punc-47::parg-2* expressing animals were obtained by injecting *parg-2(ok980); oxIs12* worms with *pAB1019* DNA at 50 ng/μl along with *Punc-25::mCherry* at 10 ng/ul and *Pmyo-2:mCherry* at 2 ng/μl as a co-injection marker. 1 kb ladder was added at 50 ng/μl as carrier. The *pAB1019* plasmid was constructed by combining Gateway plasmids encoding the *unc-47* promoter sequence, *parg-2* coding sequence (obtained from GE Dharmacon ORFeome collection), the *unc-54 UTR* sequence, and *pCFJ150*.

## Cortical axon regeneration assay

The mouse cortical neuron axon regeneration assay was performed by scrape injury of confluent cultures, as described previously (*Huebner et al., 2011*). Primary cortical cultures were established

from E17 C57BL/6 mice. Digested cells were plated on 96-well poly-D-lysine coated plates at a density of 25,000 cells per well in 200 μL of plating medium. Lentiviral particles encoding control non-targeting or PARP1 shRNA clones (Sigma) were added on DIV3 (Day In Vitro 3) as described for other shRNAs (Zou et al., 2015). On DIV8, 96-well cultures were scraped using a custom-fabricated 96-pin array and allowed to regenerate for another 72 hr before fixing with 4% paraformaldehyde. Regenerating axons in the scrape zone were visualized using an antibody against β3 tubulin (1:2000, mouse monoclonal; catalog #G712A; Promega). Growth cones were visualized by staining for F-actin using rhodamine-conjugated phalloidin (1:2000, catalog #R415, Life Technologies). Cell density was visualized using nuclear marker DAPI (0.1 μg/mL, catalog #4083, Cell Signaling Technology). Images were taken on a 10X objective in an automated high-throughput imager (ImageXpress Micro XLS, Molecular Devices) under identical conditions. Regeneration zone identification, image thresholding and quantitation were performed using an automated MATLAB script in a fully automated fashion.

### Synapse analysis

The synapse marker *hpIs3[punc-25::SYD-2::GFP; lin-15+]* was crossed onto indicated combinations of *parg(lf)* and *dlk-1(OE)* backgrounds. The dorsal cords of the resulting animals were imaged with a 40X oil objective on an UltraVIEW Vox (PerkinElmer) spinning disc confocal microscope (Nikon Ti-E Eclipse inverted scope; Hamamatsu C9100-50 camera) with Volocity software (Improvision). Images were analyzed with ImageJ.

### PARP inhibitors

All PARP inhibitors were acquired from Selleckchem. To assess the effects of PARP inhibitors on regeneration, GABA axons in L4 animals were axotomized and the worms placed on NGM plates (Brenner, 1974) containing 100 μM of the respective PARP inhibitor. Control plates were made with the same amount of DMSO as the plates containing inhibitor. Axon regeneration was scored 24 hr post-axotomy. Functional recovery was assessed by counting the number of body bends an animal made after being tapped on the nose with a platinum wire. Zero body bends is referred to as 'shrinker'.

## Acknowledgements

This work was supported by the Craig H Neilsen Foundation to ABB, by NIH R21NS082667 to MH and DMM, by NIH R33NS079306 to SMS and by the Falk Medical Research Trust to SMS.

## Additional information

### Funding

| Funder | Grant reference number | Author |
|---|---|---|
| Craig H. Neilsen Foundation | 279931 | Alexandra B Byrne |
| National Institutes of Health | R21NS082667 | David M Miller<br>Marc Hammarlund |
| National Institutes of Health | R33NS079306 | Stephen M Strittmatter |
| Falk Foundation | | Stephen M Strittmatter |

The funders had no role in study design, data collection and interpretation, or the decision to submit the work for publication.

### Author contributions

ABB, Conception and design, Acquisition of data, Analysis and interpretation of data, Drafting or revising the article; RDM, YS, Acquisition of data, Analysis and interpretation of data; SMS, DMM, MH, Conception and design, Analysis and interpretation of data, Drafting or revising the article

### Author ORCIDs

Alexandra B Byrne, http://orcid.org/0000-0002-7449-9188
Stephen M Strittmatter, http://orcid.org/0000-0001-8188-3092

David M Miller III, http://orcid.org/0000-0001-9048-873X
Marc Hammarlund, http://orcid.org/0000-0002-3068-068X

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
