## [Decision Letter]

Thank you for submitting your work entitled "Inhibiting Poly(ADP-Ribosylation) Improves Axon Regeneration" for consideration by *eLife*. Your article has been reviewed by 2 peer reviewers, and the evaluation has been overseen by Kang Shen as the Reviewing Editor and a Senior Editor.

The reviewers have discussed the reviews with one another and the Reviewing Editor has drafted this decision to help you prepare a revised submission.

Both reviewers found this paper interesting and worth publishing despite the recently published paper that is relevant to this manuscript. Both reviewers found similar strengths and weaknesses in this work.

Summary:

Byrne and colleagues present an interesting manuscript that demonstrates a role for poly-ADP-ribosylation in the regulation of axonal regeneration in *C. elegans*. They find that overexpressing DLK, which promotes regeneration, leads to the upregulation of PARGS (glycohydrolases that remove PAR [poly(ADP-ribose)]. They show that loss of PARGS inhibits regeneration. PAR is generated by PARP1-here it is shown that genetic or pharmacological loss of PARP1 enhances regeneration. This is an interesting and well-done study that makes a significant contribution to the field. However, there are a number of suggestions to improve the manuscript. For example, both reviewers found the mammalian part of the paper to be somewhat preliminary and suggested cutting it out. Both reviewers also found it necessary to include more careful interpretation and discussion of the PARG double mutant, especially in the DLK OE case. One reviewer would also like to see the figure modified to include the *parg1, 2* DKO on the same histogram as the *dlk oe, parg1,2* DKO.

Essential revisions:

1) Very recently, Langley and colleagues published a PNAS paper (PMID 26598704) that reached very similar conclusions. They showed in mammalian neurons that inhibiting PARPs improves axon regeneration, and also that axon injury induces the formation of PAR. I think two near simultaneous papers will highlight this interesting new result, and I find value in publishing the worm results. However, the Discussion section should incorporate this new paper.

2) While the results presented are compelling, there is some question about the physiological relevance of the induction of PARGS, as this was only demonstrated in the case of DLK overexpression. While the authors do show that PARGs are necessary for regeneration, they do not show that they are regulated by injury. The authors have created a very nice transcriptional reporter of PARGs that has single cell resolution. It would strengthen the paper to test whether injury activates PARG using this reporter. If PARGs are regulated by injury, this would imply that they are an active part of the injury signaling mechanism rather than merely permissive factors for regeneration.

3) The *parg1/parg2* double mutant has the same phenotype as each single mutant. While this could make sense for two components of a single pathway, that is not expected for these enzymes. Some discussion of this unexpected result is warranted.

4) There is a small figure showing some role for Parp1 in mammalian axon regeneration. This makes use of shRNA targeting *parp1*. Appropriate controls were not included. First, it should be demonstrated that the shRNA actually leads to loss of *parp1*. Second, an effort should be made to show the phenotype is not due to an off-target effect. This could be done with sh-resistent cDNA rescue. An alternative is to use two independent PARP1 inhibitors (I suggest two to get around off target drug effects). These drugs are very well validated, and would give the story a hint of translational potential. If these controls cannot be done, this small figure should be removed. The point that this pathway is conserved in mammals is well demonstrated in the Langley paper.

5) The authors claim that the DLK overexpression phenotype is partially suppressed by loss of *parg1* and *parg2*. In Figure 2, it would be nice to include the level of regeneration seen in the *parg1/2* double knockout, so that it is clear to the reader that DLK OE still leads to improved axon regeneration in the absence of *parg1/2*. Indeed, I am unconvinced that it is fair to call this suppression. Loss of *parg1/2* leads to less regeneration in both the wild type background and the DLK overexpression background. In fact, DLK overexpression leads to very robust increase in regeneration in the absence of *parg1/2* (from ~30% to ~70%). In the presence of parg, DLK overexpression leads to an increase in regeneration from ~70% to about ~90%. Obviously there is a ceiling effect so it is difficult to compare, but I don't think these results clearly show that PARG is acting downstream of DLK. These are also the results you would expect if PARG were acting in a parallel pathway. A more balanced discussion of the interpretation is warranted.

6) It is less clear on the significance of this pathway in regulating axon regeneration given the partial phenotype of *parg1/parg2* double mutant, with or without *dlk-1 OE*. With a single mammalian neuron culture experiment with shRNA knockdown, it also remains open how much this pathway plays a role in the mammalian system. Any speculation on the link between a role in axon regeneration and well established roles of PARGs and PARPs?

7) Related to #1 above, may want to do the same level of analyses in Figure 2 as in Figure 2 to dig deeper into the phenotype since at the first glance the phenotype is not strong in Figure 2. Also for Figure 2, how does *dlk-1(OE);parg-1(-);parg-2(-)* compare to *dlk-1(OE);pmk-3(-)?*

8) Any way to directly measure poly(ADP-ribose) (PAR) levels? The conclusion is on PAR levels but there is no direct evidence on this.

9) How does axon injury regulate *dlk-1 – parg* and possibly *dlk-1 – parp*? E.g., Does injury upregulate *parg-2* reporter gene expression? Is *parp* downregulated in *dlk-1 OE*? Given the experiments in cortical neurons, is PARG a transcriptional target of DLK in mammalian neurons?

10) Why pick this gene (*parg-2*)? Top hit by fold change? More detailed description of top hits from the screen? E.g., any DLK-dependent and PMK3-independent hits?

---

## [Author Response]

Essential revisions:

*1) Very recently, Langley and colleagues published a PNAS paper (PMID 26598704) that reached very similar conclusions. They showed in mammalian neurons that inhibiting PARPs improves axon regeneration, and also that axon injury induces the formation of PAR. I think two near simultaneous papers will highlight this interesting new result, and I find value in publishing the worm results. However, the Discussion section should incorporate this new paper.*

Brochier et al. found inhibiting PARP improves neurite outgrowth of murine primary cortical neurons on inhibitory substrates in vitro, PAR is upregulated in primary neuronal cultures exposed to inhibitory growth molecules in vitro, and PAR is upregulated in crushed optic nerves in vivo. We agree that these results support our finding that after axon injury, PAR regulates axon regeneration and have incorporated discussion of these results into the paper. The revised manuscript includes the following:

“in vivo, injured mammalian axons must overcome extrinsic growth inhibition to regenerate. PAR levels are upregulated in murine cortical neurons exposed to inhibitory growth molecules (myelin-associated glycoprotein, Nogo-A, Chondroitin sulfate proteoglycans) in vitro and in crushed optic nerves in vivo. […] Finally, the conservation of the role of PARP in mammalian axon regeneration may have important implications for nerve repair following injury or disease.”

*2) While the results presented are compelling, there is some question about the physiological relevance of the induction of PARGS, as this was only demonstrated in the case of DLK overexpression. While the authors do show that PARGs are necessary for regeneration, they do not show that they are regulated by injury. The authors have created a very nice transcriptional reporter of PARGs that has single cell resolution. It would strengthen the paper to test whether injury activates PARG using this reporter. If PARGs are regulated by injury, this would imply that they are an active part of the injury signaling mechanism rather than merely permissive factors for regeneration.*

Thank you for the suggestion. As suggested, we investigated PARG expression after injury using the PARG reporter. We found expression of the PARG reporter is increased in GABA neurons after injury and the increase is dependent on *dlk-1*. This experiment demonstrates PARG is an active part of the injury signaling mechanism. This experiment also identified an unexpected role for *dlk* in regulating PARG in uninjured neurons.

We have included this result as Figure 1, and modified the text as follows:

“To further test whether DLK regulates *parg* expression in GABA neurons, we built a reporter construct that expresses nuclear-localized mCherry driven by the *parg-2* promoter. […] Thus, *parg-2* expression is upregulated in injured neurons and their neighbors, and dependent on *dlk-1.*

*3) The parg1/parg2 double mutant has the same phenotype as each single mutant. While this could make sense for two components of a single pathway, that is not expected for these enzymes. Some discussion of this unexpected result is warranted.*

We agree and have included discussion of the result as follows:

” The lack of additive phenotype in the double loss of function (*parg-1(lf) parg-2(lf))* mutant suggests the two *parg* genes are not partially redundant. […] The *parg-1(lf) parg-2(lf)* mutant described here may be useful for further characterization of animals that completely lack PARG function.”

*4) There is a small figure showing some role for Parp1 in mammalian axon regeneration. This makes use of shRNA targeting parp1. Appropriate controls were not included. First, it should be demonstrated that the shRNA actually leads to loss of parp1.*

We agree and to address the question, we performed western blots on cortical neurons exposed to negative control shRNA or to two unique shRNAs that target PARP1. We found exposure to PARP1 shRNAs caused a loss of PARP1 in cortical neurons. We have included the results in Figure 2.

We included the following description in the Results and Discussion of the revised manuscript:

“To confirm the shRNA clones targeted PARP1, we performed western blots on cortical neurons exposed to negative control shRNA or to two unique shRNAs that target PARP1. […] PARP was significantly reduced in neurons exposed to either of the PARP1-targeting shRNAs compared to negative control shRNA (Figure 2). PARP levels were normalized to actin levels in each sample of neurons.

Second, an effort should be made to show the phenotype is not due to an off-target effect. This could be done with sh-resistent cDNA rescue. An alternative is to use two independent PARP1 inhibitors (I suggest two to get around off target drug effects). These drugs are very well validated, and would give the story a hint of translational potential.

We agree that we should control for off-target effects. We have two lines of evidence that demonstrate the increased regeneration seen in mammalian neurons after PARP inhibition is not due to an off-target effect: 1) Regeneration of cortical neurons is significantly increased when PARP function is inhibited by either of two unique shRNAs, 2) Regeneration of cortical neurons is significantly increased when PARP function is inhibited chemically. Together, these experiments suggest PARP is the relevant target.

These experiments are represented in Figure 2, Figure 4 and described in the Results and Discussion section of the revised manuscript. In addition, Brochier et al. independently found cortical axon outgrowth on restrictive substrates is increased when PARP is inhibited with another chemical PARP inhibitor (Brochier et al., 2015). This complementary finding is described in the last paragraph of the Results and Discussion section.

” We cultured primary cortical mouse neurons in 96-well plates (Huebner et al., 2011). […] We found that axons exposed to PARP1 shRNA regenerated significantly better than axons exposed to control shRNA (Figure 2).”

“We found that treatment with chemical PARP inhibitors after injury resulted in significantly enhanced axon regeneration in vivo in *C. elegans* GABA neurons and in vitro in murine cortical neurons (Figure 4, and Figure 4—figure supplement 2).”

*5) The authors claim that the DLK overexpression phenotype is partially suppressed by loss of parg1 and parg2. In Figure 2, it would be nice to include the level of regeneration seen in the parg1/2 double knockout, so that it is clear to the reader that DLK OE still leads to improved axon regeneration in the absence of parg1/2.*

We included the *parg1/2* double knockout data in Figure 2 (previously Figure 2) and address the question of whether PARG is downstream of DLK below. The revised manuscript includes all four genotypes in the figure (now Figure 2 and included below): wild type, *parg1/2*, DLK OE, and *parg1/2*; DLK OE.

*Indeed, I am unconvinced that it is fair to call this suppression. Loss of parg1/2 leads to less regeneration in both the wild type background and the DLK overexpression background. In fact, DLK overexpression leads to very robust increase in regeneration in the absence of parg1/2 (from ~30% to ~70%). In the presence of parg, DLK overexpression leads to an increase in regeneration from ~70% to about ~90%. Obviously there is a ceiling effect so it is difficult to compare, but I don't think these results clearly show that PARG is acting downstream of DLK. These are also the results you would expect if PARG were acting in a parallel pathway. A more balanced discussion of the interpretation is warranted.*

We have rewritten this section of the manuscript and removed the term ‘suppression’. We agree, our finding that loss of *parg1/2* reduces regeneration in both the wild type and the DLK overexpression backgrounds does not on its own show that parg is downstream of DLK. However, this finding, together with our finding that DLK overexpression increases parg expression (Figure 1), and our finding that parg expression is dependent on DLK (Figure 1), strongly suggest the PARGs act downstream of *dlk-1*. However, we cannot rule out the possibility that these pathways also have parallel functions. The revised section of the manuscript now reads:

“We assessed regeneration in animals with activated DLK signaling (*dlk-1(OE)*), but lacking both *parg-1* and *parg-2.* […] Thus, DLK-dependent regeneration depends in part on *parg-1* and *parg-2.*”

“Together, our findings suggest that regulation of PARylation is an important component of the DLK pathway role in the axon regeneration mechanism. […] Some of these functions may be mediated by other transcriptional outputs of DLK signaling (Watkins et al., 2013). Understanding these factors, as well as understanding the cellular effects of PAR on regeneration, await further study.”

*6) It is less clear on the significance of this pathway in regulating axon regeneration given the partial phenotype of parg1/parg2 double mutant, with or without dlk-1 OE.*

We agree that compared to the DLK pathway, the PARGs and PARPs have a smaller effect on regeneration. However, our work on the PARGs/PARP pathway identifies a novel mechanism, PARylation, that significantly regulates the ability of an injured axon to regenerate after injury. Further, this mechanism is distinct from DLK in that it is 1) druggable; 2) affects functional recovery after injury; and 3) does not affect synaptic morphology. Thus, the role of the PARGs and PARPs is different from DLK. In our opinion, our findings are significant because they identify PARylation as important to the cell biology of axon regeneration, they reveal an axon regeneration-specific mechanism downstream of DLK, and they show that drugs that target this pathway improve axon regeneration and functional recovery.

*With a single mammalian neuron culture experiment with shRNA knockdown, it also remains open how much this pathway plays a role in the mammalian system.*

Please see our first two responses to #4 above.

*Any speculation on the link between a role in axon regeneration and well established roles of PARGs and PARPs?*

Please see our response to #8 below.

*7) Related to #1 above, may want to do the same level of analyses in Figure 2 as in Figure 2 to dig deeper into the phenotype since at the first glance the phenotype is not strong in Figure 2.*

Figure 2 is now Figure 2. We reanalyzed the data and determined whether each axon regenerated below the midline of the worm (M-), beyond the midline (M+), at least ¾ of the distance to the dorsal cord (M++). The analysis confirmed our initial finding that the greatest difference in regenerative ability between the genotypes is in overall regeneration, in other words, axons that showed some degree of regeneration (all blue bars in Figure 2—figure supplement 1). This analysis also indicates other, more granular phenotypes that may be of interest. We added the figure to the supplemental figures (Now Figure 2—figure supplement 1).

*Also for Figure 2, how does dlk-1(OE);parg-1(-);parg-2(-) compare to dlk-1(OE);pmk-3(-)?*

As published in Hammarlund et al., 2009, 7% of *dlk-1OE; pmk-3(-)* axons regenerate. Thus, the suppression of *dlk-1OE* by *parg-1* and *parg-2* is not as strong, suggesting regulation of the *parg* genes is not the only mechanism that mediates regeneration downstream of the canonical *dlk-1* pathway. We have modified the text as follows (Results and Discussion section):

“We assessed regeneration in animals with activated DLK signaling (*dlk-1(OE)*), but lacking both *parg-1* and *parg-2.* […] Thus, DLK-dependent regeneration depends in part on *parg-1* and *parg-2.*”

*8) Any way to directly measure poly(ADP-ribose) (PAR) levels? The conclusion is on PAR levels but there is no direct evidence on this.*

Unfortunately, although PAR levels can be measured in bulk preps, we do not have a way to detect the relevant PAR levels, specifically within the GABA neurons. Our data suggests PAR levels regulate axon regeneration since PARPs and PARGs have contrasting effects on PAR levels (Bai, 2015) and on axon regeneration. In addition, Brochier et al. found PAR levels are increased in injured axons in the optic nerve (Brochier et al., 2015). We modified the text as follows:

“Since PARPs and PARGs have contrasting effects on PAR levels, NAD+ levels (Bai, 2015) and on axon regeneration, we conclude the balance between PARP and PARG function regulates axon regeneration, and present the hypothesis the PARG-PARP balance may determine axon regeneration by regulating PAR levels or by regulating NAD+ levels.”

*9) How does axon injury regulate dlk-1 – parg and possibly dlk-1 – parp? E.g., Does injury upregulate parg-2 reporter gene expression? Is parp downregulated in dlk-1 OE? Given the experiments in cortical neurons, is PARG a transcriptional target of DLK in mammalian neurons?*

As discussed in response to comment #2 above, expression of the glycohydrolase (*parg-2*) reporter gene is upregulated by injury, and expression in both injured and uninjured neurons is *dlk-1*-dependent. By contrast, expression of the polymerases (*parp-1* or *parp-2*) is not significantly changed in *dlk-1OE* axons relative to wild type axons. PARG was not identified as one of 342 genes upregulated in a DLK-dependent manner in crushed optic nerves (Watkins et al., 2013). However, the Watkins investigation was performed three days after injury and does not preclude an earlier DLK-dependent response to injury.

*10) Why pick this gene (parg-2)? Top hit by fold change? More detailed description of top hits from the screen? E.g., any DLK-dependent and PMK3-independent hits?*

*parg-2* was the fourth-most upregulated gene in the *dlk-1OE* background whose upregulation was dependent on the downstream kinase *pmk-3*. Similarly, *parg-1* was also upregulated in the *dlk-1OE* background (Figure 1). The presence of both paralogs made this gene family a particularly intriguing candidate. We did not include a description of any other candidate genes in this manuscript since including them would require a significant amount of validation, which we believe is beyond the scope of this paper. We have rewritten this section of the manuscript to describe how the *parg* genes were selected for analysis. The revised section now reads:

“RNA sequencing and analysis suggested the *parg* genes as candidates for further evaluation. […] These data suggested that regulation of PARG function might be a major effect of DLK signaling.”